# Immunomodulatory Drugs Alter the Metabolism and the Extracellular Release of Soluble Mediators by Normal Monocytes

**DOI:** 10.3390/molecules25020367

**Published:** 2020-01-16

**Authors:** Ida Marie Rundgren, Anita Ryningen, Tor Henrik Anderson Tvedt, Øystein Bruserud, Elisabeth Ersvær

**Affiliations:** 1Department of Biomedical Laboratory Scientist Education and Chemical Engineering, Faculty of Engineering and Natural Sciences, Western Norway University of Applied Sciences, 5020 Bergen, Norway; imru@hvl.no (I.M.R.); anita.ryningen@hvl.no (A.R.); elisabeth.ersver@hvl.no (E.E.); 2Department of Clinical Science, University of Bergen, 5020 Bergen, Norway; 3Section for Hematology, Department of Medicine, Haukeland University Hospital, 5021 Bergen, Norway; tor.henrik.anderson.tvedt@helse-bergen.no

**Keywords:** monocytes, immunomodulatory drugs, cell metabolism, cytokines

## Abstract

Immunomodulatory drugs (IMiDs) are used in the treatment of hematological malignancies, especially multiple myeloma. IMiDs have direct anticancer effects but also indirect effects via cancer-supporting stromal cells. Monocytes are a stromal cell subset whose metabolism is modulated by the microenvironment, and they communicate with neighboring cells through extracellular release of soluble mediators. Toll-like receptor 4 (TLR4) is then a common regulator of monocyte metabolism and mediator release. Our aim was to investigate IMiD effects on these two monocyte functions. We compared effects of thalidomide, lenalidomide, and pomalidomide on in vitro cultured normal monocytes. Cells were cultured in medium alone or activated by lipopolysaccharide (LPS), a TLR4 agonist. Metabolism was analyzed by the Seahorse XF 96 cell analyzer. Mediator release was measured as culture supernatant levels. TLR4 was a regulator of both monocyte metabolism and mediator release. All three IMiDs altered monocyte metabolism especially when cells were cultured with LPS; this effect was strongest for lenalidomide that increased glycolysis. Monocytes showed a broad soluble mediator release profile. IMiDs decreased TLR4-induced mediator release; this effect was stronger for pomalidomide than for lenalidomide and especially thalidomide. To conclude, IMiDs can alter the metabolism and cell–cell communication of normal monocytes, and despite their common molecular target these effects differ among various IMiDs.

## 1. Introduction

The immunomodulatory drugs (IMiDs) are widely used in the treatment of multiple myeloma and are also considered for the treatment of other hematological malignancies [1]. Cereblon is a common molecular target for IMiDs, and the drugs can have direct anticancer effects on malignant cells or indirect effects mediated via cancer-supporting nonmalignant cells (e.g., antiangiogenic effects) [2]. Monocytes are important both for immunoregulation and for regulation of normal and malignant hematopoiesis [3,4], and IMiD effects on monocytes may therefore be important both for their efficiency and toxicity in anticancer treatment. 

Monocytes undergo morphological, phenotypic, and functional changes in response to their metabolic microenvironment [5]. Most circulating monocytes have a classical phenotype and their metabolism can be altered by ligation of Toll-like receptor 4 (TLR4), e.g., by lipopolysaccharide (LPS) or certain metabolites (e.g., oleic acid, palmitic acid), and the downstream NFκB activation following TLR4 ligation [6]. However, very high levels of TLR4 ligands may instead induce tolerance [7]. Cholesterol, triglyceride-rich lipoproteins, low density lipoproteins (LDL), very low density lipoproteins (VLDL) and high density lipoprotein (HDL) can also modulate the monocyte phenotype through non-TLR4 mechanisms and thereby alter the balance between pro- and anti-inflammatory effects [5,6,7,8]. Finally, even dietary intake seems to influence the phenotypic characteristics of monocytes, including the balance between pro- and anti-inflammatory effects [8]. Thus, monocyte functions are modulated by the monocyte’s metabolic status/environment. 

TLR4 receptors can be activated not only by microbial products but also by endogenous ligands as described above [9,10]. TLR4 ligation can stimulate monocyte release of several cytokines [11,12] and at the same time also activate a metabolic switch towards glycolysis leading to production of metabolites that are important for the pentose–phosphate pathway, fatty acid synthesis, and amino acid metabolism [8]. This relative block in the citric acid cycle will increase the availability of citric acid and succinate with further modulation of phospholipid and cholesterol synthesis [13,14]. Thus, the balance between glycolysis and oxidative phosphorylation is not only modulated by TLR4 ligands but also by the metabolomic profile of the extracellular microenvironment [15], e.g., glutamine that both feeds into the tricarboxylic acid cycle and also acts as a regulator of TLR4 responses [13,16]. These examples further illustrate the complex overlap/crosstalk between TLR4 signaling, metabolic regulation, and extracellular mediator release. 

Monocytes can influence the development of various malignancies through direct and indirect effects [17], possibly also multiple myeloma. First, the balance between various leukocyte subsets in peripheral blood seems to have a prognostic impact, and high neutrophil- or monocyte-to-lymphocyte ratios at the time of diagnosis are associated with unfavorable clinicobiological features [18]. Secondly, monocytes can modulate the cell surface molecular profile of the myeloma cells [19]. Third, activated monocytes release cytokines that directly stimulate myeloma cell proliferation or indirectly facilitate disease progression, e.g., through increased angiogenesis [20,21]. Finally, monocyte-derived dendritic cells can present myeloma-associated antigens [22], whereas monocyte-derived myeloid-derived suppressor cells have immunosuppressive effects [23]. Thus, monocytes may influence myeloma development/progression and susceptibility to chemotherapy, but the final effects are difficult to predict. 

The immunomodulatory drugs (IMiDs) thalidomide, lenalidomide, and pomalidomide are widely used in myeloma treatment [24] and can inhibit TLR4/LPS-induced secretion of inflammatory cytokines [25]. Cereblon is a common molecular target for these drugs, but their cereblon binding as well as their clinical effects in myeloma treatment differ [2]. As described above there may be a crosstalk between TLR4 signaling, metabolic modulation, and capacity of cytokine release in normal monocytes. In this context we have compared the effects of these IMiDs on monocyte metabolism and release of soluble mediators by normal monocytes. 

As described above monocytes seem to support disease development in multiple myeloma [17], but because monocytes are a part of the bone marrow stem cell niches they may also influence the development and chemosensitivity of other hematological malignancies, e.g., through their release of leukemia-supporting cytokines [4]. IMiDs are therefore considered for the treatment of other hematological malignancies [2]. The aim of the present study was to use standardized in vitro models to investigate how various IMiDs influence important phenotypic characteristics of normal monocytes and whether these effects differ among various IMiDs that have cereblon as a common molecular target [26]. Thus, monocytes are important in physiological immunoregulation as well as in carcinogenesis, and our goal was to investigate IMiD effects on metabolic regulation and cytokine release; two functional characteristics that seem to be important are the roles of normal monocytes in both immunoregulation and carcinogenesis [8,17]. 

## 2. Results

### 2.1. TLR4/LPS Activation Alters the Balance between Glycolysis and Oxidative Phosphorylation in the Direction of Glycolysis 

Monocytes derived from 10 healthy blood donors (4 males and 6 females; median age 33 years with range 25–71 years) were incubated for four hours in medium with and without LPS; the cells were thereafter incubated for one hour without CO_2_ before cell metabolism was analyzed. We first investigated the basal mitochondrial metabolism (see Appendix A upper part) of the monocytes; the balance between oxidative phosphorylation (oxygen consumption rate, OCR) and glycolysis (i.e., extracellular acidification rate, ECAR) is presented as the OCR:ECAR ratio. The results from a representative experiment are presented in Appendix A (lower part). 

Preincubation for four hours with LPS 1 ng/mL decreased the OCR:ECAR ratio significantly, compared with the medium control (Figure 1a and Figure 2A). A similar effect on the OCR:ECAR ratio was also seen when analyzing the maximal respiration period where LPS also decreased this ratio significantly (Figure 1b and Figure 2D). Thus, LPS altered the balance between glycolysis and oxidative phosphorylation in the direction of glycolysis, but LPS did not alter the coupling efficiency (i.e., the efficiency of mitochondrial ATP production) or spare respiratory capacity (i.e., showing how much of the respiratory capacity was being used) (Figure 1c).

### 2.2. IMiDs Differ in Their Effects on the Balance between Glycolysis and Oxidative Phosphorylation; Especially Lenalidomide Increases Oxidative Phosphorylation 

We first investigated whether thalidomide, lenalidomide, and pomalidomide altered monocyte metabolism when cells from 10 healthy individuals were cultured in medium alone without TLR4/LPS stimulation. The OCR:ECAR ratio was not significantly altered by any of the IMiDs when analyzing the initial basal metabolism or the maximal respiration (Figure 1a,b). 

We then investigated the effects of the three IMiDs on the OCR:ECAR ratio in the presence of LPS/TLR4 ligation, i.e., an intervention that alters the metabolic balance in the direction of glycolysis (see above). Lenalidomide caused a significant increase in this ratio when testing the cells under basal conditions (Appendix A lower part, Figure 1a and Figure 2B), but for 8 out of the 10 tested individuals the ratio was still lower for the lenalidomide cultures than for the medium controls. The lenalidomide levels were also significantly different from the thalidomide values (Figure 2C) but did not differ significantly from pomalidomide (data not shown). Thus, lenalidomide was the only IMiD that significantly altered monocyte metabolism and decreased the relative importance of glycolysis (i.e., increased the OCR:ECAR ratio) in the presence of TLR4/LPS activation. 

We also investigated the OCR:ECAR ratio during maximal respiration (see Figure 1, Figure 2 and Figure 3). The ratio was significantly higher for lenalidomide than for pomalidomide when testing normal monocytes incubated in medium alone without LPS (Figure 2E). In the presence of LPS the IMiDs had different effects on the OCR:ECAR ratio; lenalidomide caused a highly significant increase (Figure 2F) whereas pomalidomide caused an increase of borderline significance (Figure 2G). The ratio was significantly higher for lenalidomide than for thalidomide (Figure 2H) in the presence of LPS. 

None of the IMIDs had significant effects on the coupling efficiency (i.e., the efficiency of mitochondrial ATP production, see Appendix A) (data not shown). 

### 2.3. IMiDs Have Different Effects on the Spare Respiratory Capacity of Normal Monocytes 

We first investigated the effects of the three IMiDs on the spare respiratory capacity in the absence of TLR4/LPS stimulation; none of the drugs then had any significant effect (Figure 1). We also investigated the effects of the three IMiDs on the spare respiratory capacity in the presence of TLR4/LPS stimulation (Figure 1 and Figure 3). All three IMiDs altered this capacity significantly compared with the LPS control cultures (Figure 3a–c). Lenalidomide then caused a highly significant increase (Figure 3b, *p* = 0.007), and this lenalidomide effect was significantly stronger than the increase caused by thalidomide (Figure 3d) and the decrease caused by pomalidomide (Figure 3e). Thus, the differences among IMiDs with regard to modulation of monocyte metabolism are not only reflected in OCR:ECAR ratio but also in the spare respiratory capacity. 

### 2.4. Healthy Individuals Differ in Their Spontaneous and TLR4/LPS-Induced Mediator Release

We investigated the spontaneous release of 14 soluble mediators for normal monocytes derived from 15 healthy individuals (7 males and 8 females, median age 48 years with range 23–71 years). Monocytes showed a spontaneous mediator release, but these levels were relatively low and varied between patients (Appendix A, Figure 4). 

For seven of these individuals we also investigated the release in cultures prepared in medium with LPS 1 ng/mL. As expected we observed a LPS-induced increased in soluble mediator release, and the levels did not differ significantly when we compared cultures with LPS alone and cultures with LPS + DMSO (i.e., the control cultures for the IMiD experiments) (data not shown). Thus, the TLR4/LPS-induced mediator response was maintained in the presence of the DMSO concentration used in our IMiD experiments, and the LPS response is reflected in the difference between the spontaneous levels presented in Appendix A and the levels in the DMSO containing control cultures presented in Table 1 (see also the overview in Appendix A). Finally, LPS significantly increased the levels for all soluble mediators except for CCL1 and CXCL10, and there were no significant correlations between the spontaneous release for cultures prepared in medium alone and the LPS cultures, except for IL10 (*r*-value 0.821, *p*-value 0.001) and MMP9 (*r*-value 0.621, *p*-value 0.018).

We did a hierarchical clustering analysis of the TLR4/LPS-induced mediator release for the 15 healthy individuals (Appendix A). These results are presented as the relative responses, i.e., the levels in LPS containing cultures versus the levels in control cultures prepared in medium alone. The TLR4/LPS responsiveness differed among individuals, and a strong response was seen especially for the lower six individuals (2, 10, 6, 7, 3, 1) that clustered together. We performed similar analyses for thalidomide/lenalidomide/pomalidomide containing cultures, and these analyses showed that similar variations among individuals in TLR4/LPS responsiveness (i.e., variations in absolute mediator levels persisted also in the presence of IMiDs) (data not shown).

### 2.5. Thalidomide Shows a Weak, Lenalidomide an Intermediate, and Pomalidomide a Strong Inhibitory Effect on the TLR4/LPS-Induced Release of Soluble Mediators by Normal Monocytes 

The supernatant levels of 14 soluble mediators were compared for monocyte cultures prepared in medium with DMSO (i.e., control cultures) and cultures prepared in medium with DMSO and 5 μg/mL of either thalidomide, lenalidomide, or pomalidomide (Figure 5), Appendix A, Table 1). All three IMiDs modulated TLR4/LPS-induced soluble mediator release by normal monocytes, but the effects differed among the drugs. First, thalidomide caused a significant reduction for only three mediators (CCL2, IL10, MMP9) and a highly significant effect was only seen for MMP9. Second, lenalidomide caused a significant reduction for five mediators, but a highly significant effect was only seen for CCL2. Third, pomalidomide caused a significant reduction for seven mediators and the effect was generally stronger (i.e., lower *p*-values) for pomalidomide than for the other two IMiDs (Table 1, Appendix A). A generally stronger effect (i.e., lower concentrations) by pomalidomide was also observed when comparing the absolute levels for cultures with LPS + pomalidomide versus LPS + thalidomide/lenalidomide (Appendix A). 

We also did a hierarchical clustering analysis including all the drugs and all the mediators (Figure 5). This analysis was based on the relative mediator levels in drug-containing cultures, i.e., levels in IMiD cultures relative to the level in the corresponding LPS control culture. Each of these relative responses for a mediator was next normalized to the median relative response for all three IMiDs, i.e., the median relative response of the 45 IMID responses for each soluble mediator (3 drugs times 15 individuals). It can be seen that even for mediators showing highly significant overall effects of IMiDs there was a considerable variation among individuals with regard to the effect.

Furthermore, weaker responses were often seen for the lower cluster including six patients, and all these patients showed relatively strong TLR4/LPS responses (see Appendix A). Thus, the effect of IMiDs in the presence of TLR4/LPS stimulation seemed to depend on the LPS responsiveness, and the IMiD effects were stronger for patients with a relatively weak TLR4/LPS responsiveness.

### 2.6. Lenalidomide and Pomalidomide Reduce TLR4/LPS-Induced Mediator Release by Normal Monocytes Also at Concentrations Corresponding to Their Therapeutic Serum Levels

Thalidomide was tested at a concentration corresponding to its therapeutic serum level (i.e., 5 μg/mL) in the previous experiments, whereas lenalidomide and pomalidomide in vivo levels are lower than this. We therefore tested the effect of lenalidomide 500 ng/mL and pomalidomide 100 ng/mL on the TLR4/LPS-induced mediator release for five healthy donors, and both these IMiDs could inhibit soluble mediator release even when tested at the lower concentrations corresponding to their serum levels (uncorrected *p*-value of 0.031, Wilcoxon test for paired samples).

### 2.7. Pomalidomide Inhibits Soluble Mediator Release also in the Presence of Bortezomib

The effect of the proteasomal inhibitor bortezomib was tested for five healthy blood donors (two males and three females, median age 63 years with range 21–66 years) when monocytes were cultured in medium alone or in the presence of LPS. Bortezomib had minor and divergent effects when testing monocytes cultured in medium alone (data not shown), whereas it caused an inhibition of CCL1 and CXCL10 for all five individuals when tested in the presence of LPS (Figure 6). Furthermore, we tested whether the inhibitory effects of lenalidomide and pomalidomide on TLR4/LPS-induced release by normal monocytes was maintained in the presence of bortezomib. An inhibitory effect of pomalidomide was observed for a limited number of mediators also in the presence of bortezomib, whereas lenalidomide had generally weaker and divergent effect (Figure 6). 

## 3. Discussion

Normal monocytes seem important both in immunoregulation and possibly also for carcinogenesis/leukemogegesis [1,17]. The aim of the present study was therefore to investigate whether monocyte functions are altered by IMiDs, a class of pharmacological agents that have cereblon as a common intracellular molecular target [2] and are considered for the treatment of various cancers, especially hematological malignancies [1,27,28]. Our main conclusion based on the present study is that IMiDs can alter both monocyte metabolism as well as cytokine release, and despite their common molecular target these pharmacological effects differ among IMiDs. 

Thalidomide and its derivatives lenalidomide and pomalidomide are used in the treatment of multiple myeloma [29,30,31,32,33,34]. The antimyeloma effect is probably due to both direct and indirect effects on the myeloma cells, including immunomodulation as well as anti-angiogenic, anti-inflammatory and direct antiproliferative effects [31,32,35]. These drugs are often combined with steroids and/or proteasome inhibitors [34] and can be used in frontline therapy [34]. The E3 ligase protein cereblon is a well-characterized molecular target of IMIDs [36,37], but animal studies suggest that IMiDs also have other intracellular targets [37]. The binding to cereblon modulates the stability of cereblon-interacting molecules [36] and thereby promotes apoptosis by activating caspase-8 [38]. However, the clinical evidence suggests that there are important pharmacological differences among IMiDs, and resistance against one IMiD does not exclude a response to another IMiD [39]. In the present study we investigated how IMiDs differ in their effects on normal monocytes. 

TLR4 is expressed by human monocytes [11,12,40]. LPS is a TLR4 ligand that initiates the activation of several downstream intracellular pathways in monocytes, including NFκB, extracellular signal-regulated kinases (ERK) 1 and 2, c-Jun N-terminal kinase (JNK) and p38 pathways that activate several transcription factors [11,12]. The signaling leads to metabolic modulation and a well-characterized cytokine release response including chemokines, TNFα, IL8/CXCL8, and members of the IL1 family [41]. Furthermore, TLR4 binds microbial molecules as well as several host-derived ligands, e.g., molecules derived from degradation of extracellular matrix molecules (e.g., hyaluronic acid, heparin sulphate), heat shock proteins, fibrinogen, lipoproteins, and amyloid [9,10]. Thus, TLR4 is both a cellular sensor of the extracellular microenvironment and a regulator of cellular communication. In this context we investigated the effects of IMiDs on monocyte metabolism and cytokine release in the presence of TLR4 activation.

We used highly standardized in vitro models for our comparison of pharmacological effects. The same medium was used in all studies. This medium was supplemented with inactivated fetal calf serum, and the monocytes were thereby exposed to lipids during culture. Stimulation of TLR4 by lipids or lipoproteins may at least partly explain why monocytes show spontaneous release of several soluble mediators during culture, even in the absence of TLR4 stimulation [42]. 

Our monocyte populations had a purity exceeding 83%, and a major part of the contaminating cells were small lymphocytes. Furthermore, we cultured the cell for only 24 h, whereas lymphocyte activation will often require a longer incubation time for cytokine release, even during mitogenic stimulation [43]. For these reasons we regard our cytokine responses to be monocyte responses. 

Thalidomide (molecular weight 358), lenalidomide (359), and pomalidomide (372) were tested at a concentration of 5.0 μg/mL, and due to these minor differences in molecular weight their molar levels in the cultures should in our opinion be regarded as comparable. However, despite comparable molar levels the IMiDs differed in their pharmacological effects; lenalidomide showed a stronger effect on monocyte metabolism, whereas pomalidomide had the most significant effect on soluble mediator release. This last difference in their effects on cytokine release is probably not caused by different effects on monocyte viability; if so, one would expect all mediators to be affected and not only certain mediators, as we observed. 

The concentration of 5 μg/mL was chosen because this is the systemic level reached during thalidomide treatment of myeloma patients [44]. However, the systemic levels of lenalidomide and pomalidomide are usually lower, and for this reason we also tested lenalidomide and pomalidomide at levels corresponding to their systemic levels reached during myeloma treatment (i.e., 500 and 100 ng/mL, respectively) [45,46,47]. These experiments showed that both lenalidomide and pomalidomide reduced monocyte release of soluble mediators also when tested at these lower concentrations. 

Our studies showed that LPS as expected altered monocyte metabolism in the direction of glycolysis [41]. IMiDs could further modulate the metabolism of normal monocytes in the presence of TLR/LPS stimulation, and lenalidomide had the strongest effects, with a significant increase in the OCR:ECAR ratio (i.e., increased importance of glycolysis) and an increased spare respiratory capacity. These observations were only made in the presence of TLR4/LPS stimulation but not when cells were cultured in medium alone, an observation demonstrating that the differences between IMiDs with regard to metabolic modulation depend on the biological context. Finally, even though lenalidomide seems to have the strongest effect on metabolism, pomalidomide had the strongest effect on the extracellular release of soluble mediators. Thus, the IMiD effects on these phenotypic characteristics are probably mediated at least partly through different molecular mechanisms. 

There was a considerable variation among individuals with regard to the levels of the various soluble mediators, and a possible explanation for this could be immunogenetic differences between cell donors. Single nucleotide polymorphisms (SNPs) exist both for NKκB and TLR4 signaling [48,49,50,51,52,53], and certain SNPs seem to be clinically relevant and are associated with differences in regulation of monocyte activation. First, certain polymorphisms have been associated with cancer risk, and this is possibly due to a genotypic influence on intracellular signaling [50,52]. Second, the risk and/or severity of infections have also been associated with such genetic variants [49,53]. Finally, immunogenetic characteristics may be important for the role of monocytes in the development of inflammations [51,53]. Both clinical and experimental studies suggest that differences in monocyte cytokine responsiveness are important for the associations between immunogenetic differences and severity of infections [53], but differences in TLR4 expression levels may also contribute [48]. In this context it is not surprising that we detect a considerable variation among healthy individuals in the cytokine responsiveness of normal monocytes to TLR4 ligation. 

Proteasome inhibitors are also widely used in the treatment of multiple myeloma [27]. These drugs inhibit the NFκB pathway, one of the pathways also activated by TLR4 [9], and may thereby be able to reduce the release of NFκB-regulated soluble mediators [54]. Proteasome inhibitors and IMiDs can be combined in myeloma treatment [27]. The proteasome inhibitor bortezomib had relatively weak and divergent effects on the mediator release by normal monocytes and inhibited TLR4/LPS-induced release only for a few mediators. However, bortezomib seemed to modulate the inhibitory effects of IMiDs on the mediator release, and this was true especially for lenalidomide, whereas several effects of pomalidomide were maintained also in the presence of bortezomib. 

Infections are a major cause of morbidity and mortality in myeloma patients, and immunosuppressive effects of IMiDs may contribute to the risk of infections [55]. Advanced myeloma is associated with dysregulation of several immunocompetent cells [29], including monocytes [18,56,57,58,59,60,61,62]. The effects of IMiDs on monocyte mediator release will influence communication between immunocompetent cells and possibly contribute to the risk of infections. Myeloma patients receiving antimyeloma therapy seem to have a quantitative monocyte defect [63] and our present results suggest that these patient also have a qualitative effect, but in our opinion the myeloma-associated B-cell defect is probably most important for the increased risk of infections [64].

Our present in vitro studies suggest that various IMiDs alter the metabolic regulation and the immunomodulatory functions of normal monocytes, but for several reasons our results should be interpreted with great care. First, additional clinical studies are needed to clarify whether these effects are relevant in vivo. Second, normal monocytes consist of different subsets with different immunoregulatory functions [17,65], and it is not known whether IMiDs have similar effects on different monocyte subsets. Third, it is not known whether circulating normal monocytes derived from cancer patients, especially allotransplant recipients [63,65], show similar pharmacological effects compared with monocytes from healthy individuals, or whether the IMiD effects are similar for young/middle-aged/elderly patients. Finally, we have only investigated IMiDs in combination with the proteasome inhibitor bortezomib; additional combinations used for treatment of multiple myeloma [27] as well as for other hematological malignancies [28] also need to be investigated. 

Based on our present experimental studies we conclude that IMiDs modulate metabolism and communication of normal monocytes, and despite their common molecular target [2] these effects differ among IMiDs. These pharmacological in vivo effects may be relevant for immunoregulatory as well as cancer-supporting effects of normal monocytes [1,17], and future clinical studies should therefore try to clarify whether these effects are also important in vivo. 

## 4. Materials and Methods 

### 4.1. Cell Donors

Normal monocytes were derived from healthy blood donors. In accordance with the approved routines at the Blood Bank, Haukeland University Hospital peripheral venous blood samples were donated after written informed consent. The project was approved by the Regional Ethics Committee (REK VEST 2013/635, 2017/305).

### 4.2. Reagents

A stock solution of lipopolysaccharide (LPS) from *Escherichia coli* (#L2654-1MG; Merck KGaA, Darmstadt, Germany) was dissolved in medium (1 mg/mL) and stored at −80 °C. LPS was used at a concentration of 1 ng/mL based on titration experiments using monocytes in the Seahorse assay. Bortezomib (#5043140001; Merck KGaA) was dissolved in medium and used at a final concentration of 25 nM; this concentration can inhibit in vitro constitutive chemokine release by myeloid cells [54]. Stock solutions of thalidomide 12 µg/mL (#14610), lenalidomide 16 µg/mL (#14643), and pomalidomide 15 µg/mL (#19877; all from Cayman Chemicals, Ann Abor, MI) were prepared in DMSO (D2650-5X5ML, Merck KGaA), aliquoted, and stored at −80 °C. DMSO reached a final concentration of 0.55 mg/mL (corresponding to 0.055%) in the experiments. The molecular weights for the IMiDs are thalidomide (C13H10N2O4) 258, lenalidomide (C13H13N3O3) 259, and pomalidomide (C13H11N3O4) 273 (see Appendix A). The IMiDs were used at a final concentration of 5 µg/mL; this concentration corresponds to the systemic levels reached in vivo during thalidomide treatment of myeloma patients [44] (for molar concentrations, see Appendix A). 

### 4.3. Preparation of Enriched Normal Monocytes

Monocytes were isolated from buffy coats that were diluted 1:1 with phosphate-buffered saline (PBS), and peripheral blood mononuclear cells (PBMC) were then isolated by density gradient separation (Lymphoprep™, NycoMed, Oslo Norway; density 1.077 g/mL; centrifugation 800 G/30 min). The PBMC were washed twice in PBS, resuspended in 5 mL RPMI 1640 medium (#R7509, Merck KGaA) and overlaid 4 mL Percoll solution (P4937, Merck KGaA) [66]. After centrifugation (500 G, 30 min, room temperature) monocytes were harvested, washed, resuspended in 30 mL PBS and counted by a TC20™ Automated Cell Counter (BIO-RAD, Oslo, Norway). 

The cells were thereafter centrifuged and resuspended (10^7^ cells/40 µL in the recommended buffer for the Pan Human Monocyte Isolation Kit (#130-096-537; MACS Miltenyi Biotec, Bergisch Gladbach, Germany) and the separation procedure performed strictly according to the manufacturer’s recommendations. Briefly, the LS column (#130-042-401, MACS Miltenyi Biotec), and the 30 µm preseparation filter (#130-041-407, MACS Miltenyi Biotec) were prepared according to the instructions. Cells were incubated with the FcR-block and Pan Monocyte Biotin–Antibody Cocktail for 5 min at a concentration of 10 µl per 10^7^ cells; an additional 40 µL of buffer per 10^7^ cells was thereafter added, followed by 20 µl Anti-Biotin MicroBeads per 10^7^ cells. The cell suspension was thereafter incubated for 10 min before buffer was added, the cells were centrifuged and resuspended in buffer at a concentration of 10^8^ cells/0.5 mL. Thereafter 0.5 mL of the cell suspension was added to the pre-filter and the suspension was run through the column and collected. The column was subsequently washed with 3 mL buffer that was collected together with the monocytes. The collected cells were centrifuged and reconstituted in 50 mL PBS before counting. Flow cytometric analysis of isolated cell populations showed at least 83% monocytes (median 89.5%, range 83.4–95.3%); the majority of contaminating cells being small lymphocytes. The monocytes included a median of 84% classical (range 76–90%), 5% (range 4.1–9.3%) intermediate, and 9% (range 4.8–18.4%) nonclassical monocytes. The monocytes were centrifuged and resuspended in RPMI 1640 supplemented with 10% fetal calf serum (FCS) and penicillin/streptomycin (1.25 x 10^6^ cells/mL), and 200 µl were distributed to each culture well. After 30 min of preincubation at 37 °C the cells were used in the experiments.

### 4.4. Analysis of Monocyte Metabolism

*In vitro culture of enriched monocytes.* Monocytes were incubated for four hours in medium with and without IMIDs/LPS before the medium was changed. Thereafter the cells were incubated for one hour without CO_2_ before analysis by the Seahorse XF 96 cell analyzer. No significant difference was detected for control cultures (with or without LPS) with and without DMSO. Pilot experiments compared the effects of IMiDs after 2, 4, and 24 h of in vitro culture; the effects of IMiDs were strongest after four hours and this incubation time was used for all experiments. 

*Extracellular flux assays.* The XF Mito Stress Test Kit (#103,325-100 and #103,015-100; Agilent Technologies, Inc., CA) was used strictly according to the manufacturer’s instructions. Briefly, assay medium was prepared by supplementing XF Base Medium minimal DMED (#102,353-100; Agilent Technologies) with glucose 10 mM (#103,577-100, Agilent Technologies), pyruvate 1 mM (S8,636-100 mL, Merck KGaA), and glutamine at 2 mM (#103,575, Agilent Technologies); the pH was adjusted to 7.4 with NaOH. Cells were washed twice in assay medium before being resuspended in 180 µL medium and incubated (humidified atmosphere, 37 °C, without CO_2_) for 60 min. The cartridges that had been hydrated according to manufacturer instructions one day earlier, were prepared with the drugs for injections, i.e., oligomycin 2 µM, carbonyl cyanide-p-trifluoromethox-yphenyl-hydrazon (FFCP, a protonophore) 1 µM, and rotenone/antimycin A 0.5 µM (all solutions prepared in assay medium). The extracellular flux assays were performed; subsequently the assay medium was discarded and the plate stored at −80 °C for at least 24 h before the amount of protein was measured (see below). All assays were prepared with 6–8 parallels. 

The oxygen consumption rate (OCR) and extracellular acidification rate (ECAR) were estimated by a Seahorse XF 96 cell analyzer. The spare reservoir capacity and coupling efficiency was calculated by XF Cell Mito Stress Test Excel Template. The definitions of the parameters used in the metabolic analyses can be seen in Appendix A (upper part). 

*Estimation of total cell proteins.* The Pierce™ BCA Protein Assay Kit (#23,225; Thermo Scientific™, Waltham, MA, USA) was used for normalization of data and performed strictly accordingly to manufacturer’s instructions. A total of 10 µl of standard samples was added to each well together with 200 µl of working reagent. After 30 s on a plate shaker and 30 min incubation 150 µl was transferred to a new plate and read at 595 nm by a iMark™ Microplate Absorbance Reader (Bio-Rad laboratories, Oslo, Norway). 

### 4.5. Analysis of TLR4/LPS Induced Cytokine Release

Enriched normal monocytes were cultured in RPMI 1640 medium (#R8758-1L; Merck KGaA) supplemented with 10% inactivated fetal calf serum (#S181B-500; Biowest, Nuaillé, France) and penicillin-streptomycin (#MS00AO100H; Biowest). Cultures were prepared in Agilent Seahorse XF95 cell culture microtiter (96-well) plates with each well containing 250,000 monocytes in 200 µl medium. LPS 1 ng/mL was added together with the drugs, cultures were thereafter incubated (37 °C, humidified atmosphere, 5% CO_2_) for 24 h before supernatants were harvested. The supernatants were stored frozen at −20 °C until analyzed. 

Supernatant levels of soluble mediators were determined by Luminex high performance assays (Biotechne, Abingdon, UK) and included CCL1, CCL7, CXCL10, IL-8, MMP-9, CCL2, CCL4, CXCL1, IL-1β, and IL-10. ELISA assays (Biotechne) were used to determine levels of CCL3, IL1RA, IL6, and TNF-α. All assays were performed strictly accordingly to the manufacturer’s instructions; the assays were performed in duplicates and differences between duplicates were generally <10%.

### 4.6. Statistical Analysis 

The Wilcoxon rank sum test and the Wilcoxon test for paired samples were used for statistical analyses and *p*-values < 0.05 were regarded as statistically significant. 

## Figures and Tables

**Figure 1 molecules-25-00367-f001:**
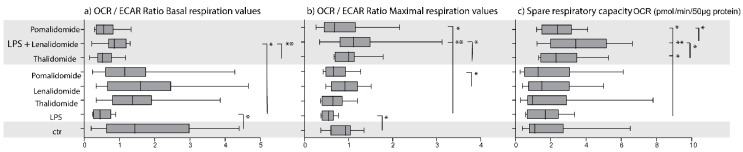
Analysis of monocyte metabolism using the XF Mito Stress Test assay and the Seahorse XF 96 cell analyzer; a summary of the overall results. Normal monocytes were cultured in medium alone or in the presence of LPS (lipopolysaccharide) 1 ng/mL, or IMiDs (immunomodulatory drugs) 5 μg/mL for four hours before the oxygen consumption rate (OCR) and extracellular acidification rate (ECAR) were determined. Each of the figures presents the OCR:ECAR ratio (diagrams (**a**,**b**)) or the spare respiratory capacity (diagram (**c**)) for cultures prepared in (from the bottom to the top of each diagram) (i) medium alone (ctr), (ii) LPS control, (iii) each of the IMiDs—thalidomide, lenalidomide, or pomalidomide—in medium alone without LPS; and (iv) LPS in combination with either thalidomide, lenalidomide or pomalidomide. The figures show the results for (**a**) the OCR:ECAR ratio during the initial period of basal incubation; (**b**) the period of maximal respiration after addition of FCCP (Carbonyl cyanide-4 (trifluoromethoxy) phenylhydrazone; an uncoupling agent); and (**c**) the calculated spare respiratory capacity. The figure summarizes the results for 10 independent experiments testing monocytes from 10 healthy blood donors. The results are presented as median, box (i.e., the 25–75 percentiles) and whiskers (the 5–95 percentiles). The Wilcoxon test for paired samples was used for the statistical comparisons, and *p*-values < 0.05 were regarded as statistically significant. The significant comparisons are indicated in the figure (* *p* < 0.05, ** *p* < 0.01).

**Figure 2 molecules-25-00367-f002:**
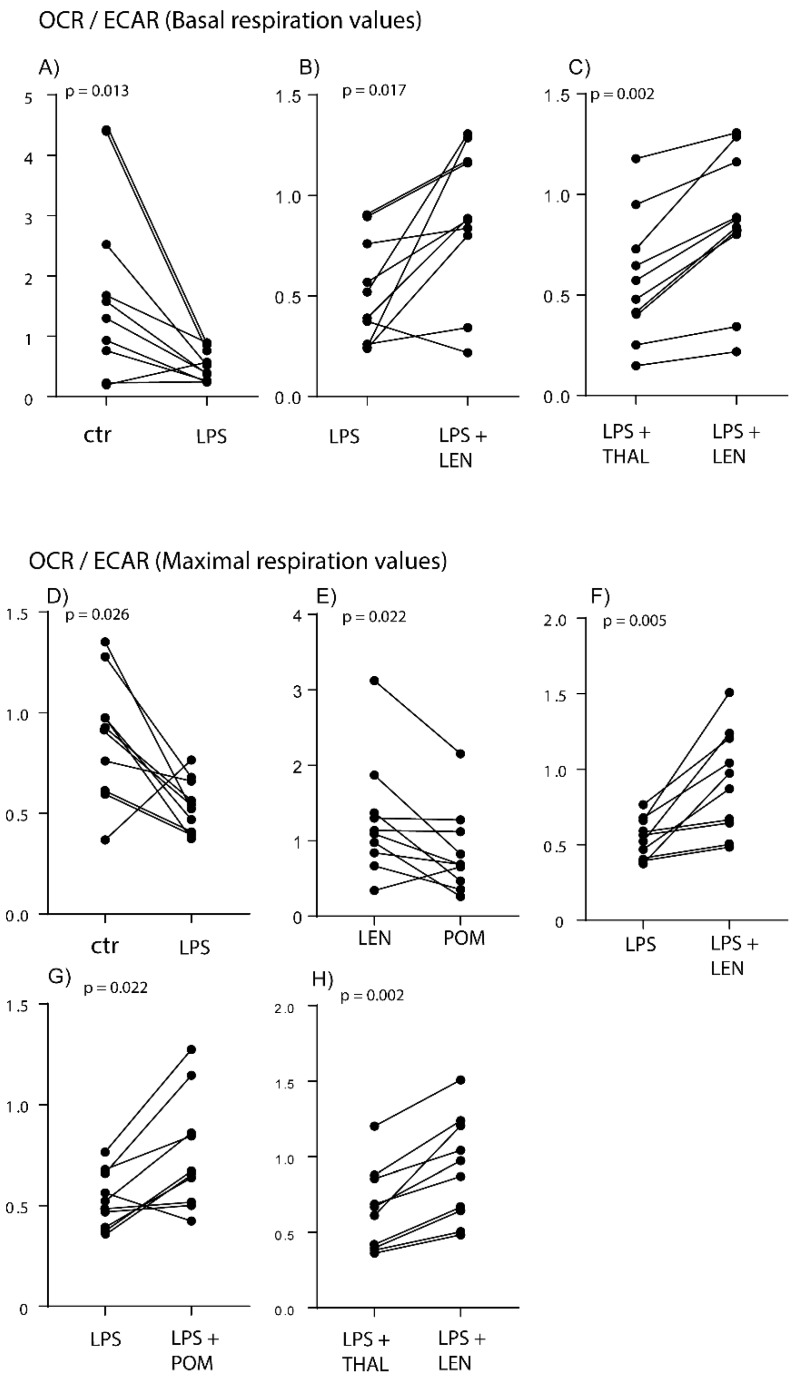
The effects of IMiDs on the metabolism of normal monocytes; a presentation of statistically significant effects of LPS and IMiDs on the OCR:ECAR ratio, (**A**–**C**) is OCR:ECAR ratio at basal levels and (**D**–**H**) at maximal respiration. Normal monocytes derived from 10 different healthy individuals were cultured with and without LPS 1 ng/mL and/or with and without IMiDs 5 μg/mL in the presence of LPS. Metabolism was analyzed using the XF Mito Stress Test assay and the Seahorse XF 96 cell analyzer. The figure presents the results for all statistically significant comparisons. The culture conditions that were compared in each of the statistical analyses/diagrams are indicated on the *x*-axis, the OCR:ECAR ratio is indicated on the *y*-axis and the corresponding *p*-values from the statistical analyses (Wilcoxon test for paired samples) are indicated at the top of each diagram. We present the results from analysis of basal (**A**–**C**) and maximal respiration (**D**–**H**). Each diagram compares the results for (i) cultures prepared with medium alone versus LPS (**A**,**D**); (ii) cultures prepared with LPS without and with an IMiD (**B**,**F**,**G**), and (iii) cultures prepared with two different IMiDs but otherwise with similar culture conditions (**E**,**H**).

**Figure 3 molecules-25-00367-f003:**
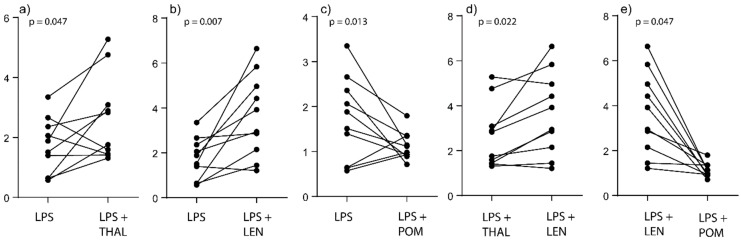
The effect of IMIDs on the metabolism of normal monocytes; analysis of the spare respiratory capacity. Normal monocytes derived from 10 healthy individuals were cultured with LPS 1 ng/mL alone or in combination with an IMiD (5 μg/mL). Monocyte metabolism was analyzed using the XF Mito Stress Test assay and the Seahorse XF 96 cell analyzer. The figure presents the results for all statistically significant comparisons. The cultures compared in each of the statistical analyses/diagrams are indicated on the *x*-axis, the spare respiratory capacity is indicated on the *y*-axis (pmol/min/50 μg protein) and the *p*-value for the statistical analysis (Wilcoxon test for paired samples) is indicated at the top of each diagram. We present the results for LPS stimulated cultures with and without an IMiD (**a–d**) and with two different IMiDs (**e**).

**Figure 4 molecules-25-00367-f004:**
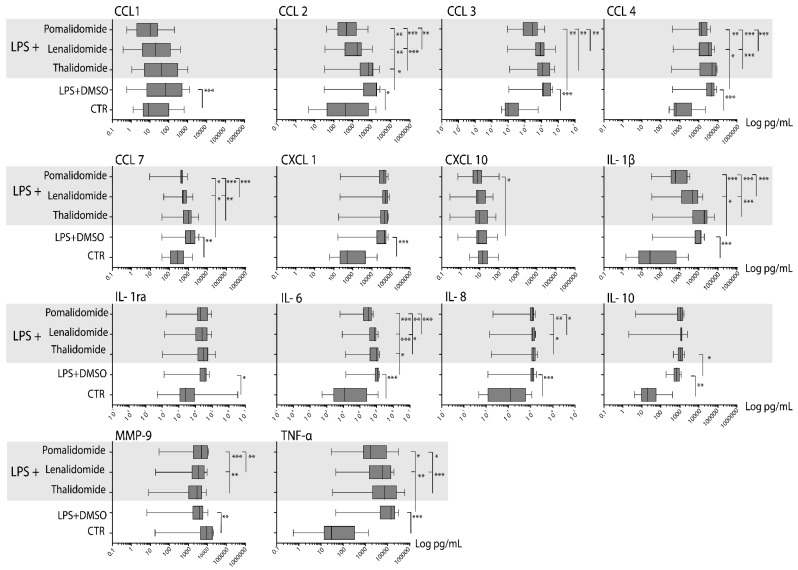
Analysis of monocyte cytokine release during in vitro culture; a summary of the overall results. Normal monocytes were cultured in medium alone or in the presence of LPS 1 ng/mL, or IMiD 5 μg/mL for 24 h before supernatants were harvested and the supernatant levels of the 14 soluble mediators determined. Each of the diagrams/figures present the level for cultures prepared in (from the bottom to the top of the figure) (i) medium alone (ctr), (ii) LPS + DMSO alone, (iii) each of the IMiDs—thalidomide, lenalidomide or pomalidomide—in combination with either thalidomide, lenalidomide, or pomalidomide. The diagrams show the results for each individual soluble mediator. The results are presented as the median, box (i.e., 25–75 percentiles), and whiskers (5–95 percentiles). The Wilcoxon test for paired samples was used for all statistical comparisons, and *p*-values < 0.05 were regarded as statistically significant. All statistically significant comparisons are indicated in the figure (* 0.01 < *p* < 0.05, ** *p* < 0.01, *** *p* < 0.001).

**Figure 5 molecules-25-00367-f005:**
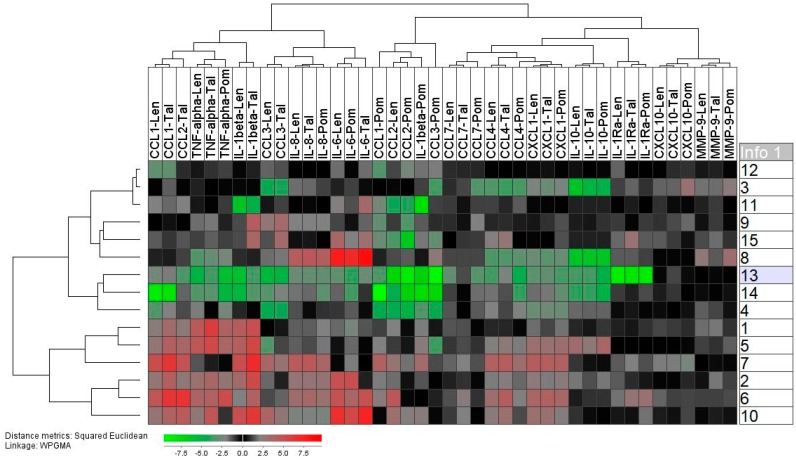
The effects of IMiDs on TLR4/LPS-stimulated cytokine release by normal monocytes; a hierarchical clustering analysis of the overall IMiD effects. The analysis was based on the relative mediator level, i.e., level in IMiD-containing LPS cultures relative to the level in the corresponding LPS stimulated IMiD-free control. Before the clustering analysis all relative responses for each mediator were normalized to the median relative response for all three IMiDs, i.e., they were normalized to the median relative response of all 45 IMID responses for each mediator (3 IMiDs, 15 individuals). Thus, green color thus means a relatively strong inhibitory effect. It can be seen that even for mediators showing highly significant effects of IMiDs there was a considerable variation among individuals with regard to the effects of IMiDs.

**Figure 6 molecules-25-00367-f006:**
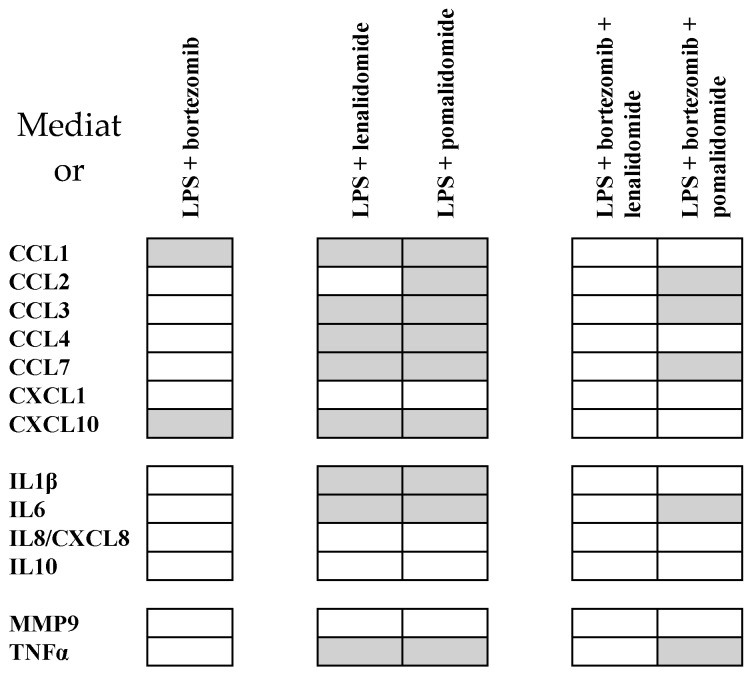
The effects of TLR4/LPS, bortezomib, and IMiDs on the release of soluble mediators by normal monocytes; a summary of the overall results for five healthy blood donors. We compared the mediator levels for (i) cultures prepared in medium with and without LPS 1 ng/mL; and (ii) DMSO containing control cultures with LPS alone versus cultures with LPS together with either lenalidomide 500 ng/mL or pomalidomide 100 ng/mL (i.e., concentrations corresponding to the serum levels reached during myeloma treatment); and (iii) cultures with LPS + bortezomib versus cultures with LPS and bortezomib together with either lenalidomide or pomalidomide. The comparisons showing a similar decrease (i.e., at least 25% reduction) in the soluble mediator level for all five individuals included in the comparisons are indicated in grey; a decrease for all five individuals corresponds to an uncorrected *p*-value of 0.031 when using the Wilcoxon test for paired samples.

**Table 1 molecules-25-00367-t001:** The effects of thalidomide, lenalidomide, and pomalidomide on TLR (Toll like receptor) 4 -induced release of soluble mediators by normal monocytes derived from healthy individuals. The data are presented as the median and range of the levels detected in culture supernatants. Fifteen healthy individuals were examined. The Wilcoxon test for paired samples was used for the statistical analyses (n.s., not significant). Unless otherwise stated all concentrations are given as pg/mL.

Mediator Classification	TLR4-Induced Release in Control Cultures	TLR4-Induced Release in the Presence of Thalidomide	*p*-Value	TLR4-Induced Release in the Presence of Lenalidomide	*p*-Value	TLR4-Induced Release in the Presence of Pomalidomide	*p*-Value
**Chemokines**							
CCL1	63.8 (0.6–1229)	40.7 (1.1–994)	n.s.	22.2 (0.4–406)	n.s.	10.4 (0.6–191)	n.s
CCL2	11,479 (30.3–26,405)	6751 (32.4–24,724)	0.036	1816 (34.5–10,349	0.003	461 (40.8–6173)	0.001
CCL3	973 (350–62,216)	198,871 (1175–566,384)	n.s.	84,338 (919–652,373)	n.s.	26,298 (878–150,656)	n.s
CCL4	43,275 (409–86,040)	48,452 (359–90,664)	n.s.	30,375 (437–66,560)	0.031	13,308 (394–31,896)	0.003
CCL7	1274 (39,7–3571)	1088 (39.8–3473)	n.s.	524 (50.7–1724)	0.011	410 (9.1–485)	0.003
CXCL1	46,239 (161–70,248)	47,546 (177–76,381)	n.s.	56.508 (214–83,477)	n.s.	43,350 (218–71,720)	n.s
CXCL10	8.3 (<0.1–86.3)	9.3 (<0.1–69.7)	n.s.	8.2 (<0.1–49.1)	n.s.	7.5 (<0.1–102)	0.019
**Interleukins**							
IL1β	13,578 (34,9–19,091)	20,310 (37.4–66,422)	n.s.	6363 (33.4–15,877)	0.02	580 (31.8–2442)	0.001
IL1RA	97,396 (130–66,232)	29,043 (102–158,380)		23,839 (137–91,545)	n.s.	19,870 (168–93,618)	n.s
IL6	117,615 (2928–104.657)	117,722 (1324–60,904)		77,344 (865–125,094)	0.0125	32,080 (615–61,886)	0.0007
IL8/CXCL8	123,383 (2692–176,743)	134,328 (1697–199,724)	n.s.	135,616 (1428–165,045)	n.s.	129,439 (1988–161,864)	n.s
IL10	931 (<2.1–2500)	1255 (<2.1–1743)	0.038	1082 (<2.1–1573)	n.s.	1092 (<2.1–1573)	n.s
**Other mediators**							
MMP9	3943 (50–10,680)	2862 (8.2–8639)	n.s.	2895 (18.5–9801)	n.s.	4856 (28.3–11,681)	n.s
TNFα	17,983 (46–21,758)	7366 (33–60,688)	0.0018	5997 (46–19,516)	n.s.	1709 (30–31,550)	0.0268

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
