# Peer review of "Immunomodulatory Drugs Alter the Metabolism and the Extracellular Release of Soluble Mediators by Normal Monocytes"

_molecules, 2020, doi:10.3390/molecules25020367_

Round 1
Reviewer 1 Report
The manuscript submitted as Article to Molecules titled
"Immunomodulatory drugs alter the metabolism and the extracellular release of soluble mediators by normal monocytes" describes the reduction of TLR4-mediators release due to immunomodulatory drugs.
The research was well-conducted and results were clearly presented. Methods are well written and the reported data were statistically significative. The manuscript was well written, with a good English and suitable for publication in Molecules after minor revisions.
My minor comments are:
Why did not you use the molar concentration in your experiments?
Please rewrite "ml" with "mL".
Add a figure with the molecular structures of the used drugs.
Reviewer 2 Report
The search for drugs used in treatment of other hematological malignancies is a very serious research task. The study authors used highly standardized in vitro models for comparison of pharmacological effects. The same medium was used in all studies, which confirms that the experiment model was planned very correctly. The methodology and description of the results obtained is understandable, especially Table 1, which presents the effects of thalidomide, lenalidomide and pomalidomide on TLR4 induced release of soluble mediators by normal monocytes derived from healthy individuals. The statistical analysis presented confirms the validity of the undertaken study.
However, the work requires correction, first of all there are no conclusions, each work has its clearly stated goal, the results obtained and it is necessary to present the conclusions. The presented work lacks a clearly formulated, separated goal and conclusions.
Secondly, it is necessary to present the weaknesses of the presented study, what else can be done or could not be achieved in the presented experiment.
Reviewer 3 Report
Manuscript entitled “Immunomodulatory drugs alter the metabolism and the extracellular release of soluble mediators by normal monocytes”. This manuscript from Rundgren IM and colleagues were trying to identify the effect of Immunomodulatory drugs (IMiDs) on monocyte metabolism and cell cell communication using F Mito Stress Test assay and the Seahorse XF 96 cell analyzer. Authors compared effects of thalidomide, lenalidomide and pomalidomide on in vitro cultured normal monocytes. This finding indicates that IMiDs modulate metabolism and communication of normal monocytes. Figure 4 image quality not good, replace with good resolution.
